# Iturin A Rescued STb-R-Induced Pork Skeletal Muscle Growth Restriction through the Hypothalamic-Pituitary-mTORC1 Growth Axis

**DOI:** 10.3390/ani12121568

**Published:** 2022-06-17

**Authors:** Mao Ye, Zhenhua Liu, Chunqi Gao, Huichao Yan, Xiuqi Wang, Liufa Wen, Chenglong Jin

**Affiliations:** College of Animal Science, South China Agricultural University, Guangdong Provincial Key Laboratory of Animal Nutrition Control, National Engineering Research Center for Breeding Swine Industry, Guangzhou 510642, China; 19854815506@stu.scau.edu.cn (M.Y.); liuzh521@stu.scau.edu.cn (Z.L.); cqgao@scau.edu.cn (C.G.); yanhc@scau.edu.cn (H.Y.); xqwang@scau.edu.cn (X.W.)

**Keywords:** Iturin A, growth axis, mTORC1, skeletal muscle growth, pig

## Abstract

**Simple Summary:**

Since antimicrobial peptides (AMPs) possess broad-spectrum antimicrobial activity and low drug resistance, their development as antibiotic alternatives are of interest. Previous studies reported that the performance of pigs was improved by AMPs through the protection of the intestine. Nevertheless, direct evidence linking AMPs and skeletal muscle growth is needed. In the current work, Iturin A as a novel antimicrobial peptide was selected to fight engineered STb-Rosetta Escherichia coli (STb-R) to measure pork skeletal muscle growth and further mechanisms. We found the restrictions of body weight gain and skeletal muscle growth caused by STb-R could be rescued after Iturin A gavage, and this was mediated by the hypothalamic-pituitary-Mtorc1 growth axis.

**Abstract:**

The engineered STb-Rosetta *Escherichia coli* (STb-R) was designed to investigate the effects of Iturin A on the skeletal muscle growth of weaned piglets. A total of 28 piglets were randomly divided into 4 groups (7 piglets per group): the control group (100 mL PBS), the Iturin A group (100 mL 320 mg/kg body weight (BW) Iturin A), the STb-R group (100 mL 1 × 10^10^ CFU/mL STb-R), and the Iturin A + STb-R group (100 mL 320 mg/kg BW Iturin A + 1 × 10^10^ CFU/mL STb-R). Compared with the control, STb-R-reduced body weight gain were rescued by Iturin A. The *semimembranosus* muscle weight recovered to normal level in the Iturin A + STb-R group. The level of relevant genes of the growth axis were elevated by Iturin A, including *GHRH* in the hypothalamus, *GHRHR* and *GH* in the pituitary, and *GHR*, *IGF-1* and *IGF-1R* in the *semimembranosus* muscle. Moreover, Iturin A increased the mean fiber area and the number of proliferating cells in the *semimembranosus* muscle, which were decreased by STb-R. Additionally, the mTORC1 pathway was reactivated by Iturin A to relieve the suppression of STb-R. Collectively, the hypothalamic-pituitary growth axis-mediated Iturin A reactivated the mTORC1 pathway to rescue STb-R-restricted pork skeletal muscle growth.

## 1. Introduction

In recent years, antibiotics have been gradually prohibited in the pig industry, which is beneficial to human life and health [1,2]. However, many harmful bacteria hinder pig growth and reduce production efficiency [3]. This is a new and important challenge for the field of animal nutritional regulation. Therefore, it is urgent to develop antibiotic alternatives (ASs) to protect animal health. Antimicrobial peptides (AMPs) are recognized as the strongest potential ASs and have been reported to improve animal performance [4].

Iturin A is an antimicrobial peptide secreted by *Bacillus subtilis* that consists of a hydrophobic fatty acid chain and a hydrophilic peptide [5,6]. This specific structure allows it to efficiently disrupt the cell membrane of target cells by interacting with phospholipids, making it difficult for bacteria to develop resistance [7,8]. As a result, Iturin A has potent antifungal and antimicrobial activity against various pathogens, such as *Aspergillus flavus* [9] and *Candida albicans* [10]. However, whether it can be used as a novel AS to promote piglet growth is unknown, and the regulatory mechanism still needs to be explored.

The hypothalamic-pituitary growth axis is an important hub in the regulation of animal growth and is involved in digestive processing, protein synthesis, and many other processes by secreting various hormones [11]. In skeletal muscle, insulin-like growth factor 1 (IGF-1), which is mainly regulated by the hypothalamic-pituitary growth axis, binds to its receptor insulin-like growth factor 1 receptor (IGF-1R) to activate downstream signaling cascades to promote muscle growth and maintain muscle mass [12,13]. Therefore, IGF-1 and IGF-1R are regarded as important biomarkers of animal health and fitness. The mammalian target of rapamycin complex 1 (mTORC1) signaling cascades, a central pathway regulating protein synthesis, have also been shown to be one of the key downstream pathways in response to IGF-1 [14]. Related studies have shown that the mTORC1 pathway enhances the proliferation and differentiation of satellite cells and accelerates protein synthesis to enhance myofiber hypertrophy [14,15]. However, the interaction between Iturin A and the IGF-1-mTORC1 growth axis has not been reported.

In the current study, we used engineered STb-Rosetta *Escherichia coli* (STb-R) as the pathogen to investigate the effects of Iturin A on the skeletal muscle growth of weaned piglets and its possible mechanisms to provide more evidence for its application in animal husbandry. In detail, we evaluated the growth of piglets and selected semimembranosus muscles to detect the mean fiber number and the mean fiber CSA. In addition, the levels of components of the growth axis and mTORC1 pathway were further detected to explore the relationship between Iturin A and skeletal muscle growth. Overall, our results show that Iturin A reactivated the mTORC1 pathway via the growth axis to rescue STb-R-restricted skeletal muscle growth.

## 2. Materials and Methods

All animal procedures were performed according to the Guidelines for the Care and Use of Laboratory Animals of South China Agricultural University (Guangzhou, China).

### 2.1. Animals and Treatments

A total of twenty-eight 28-day-old, healthy Duroc × Landrace × Large White piglets (Weaned at 25 days of age) of similar weights (Half sibs, BW = 9.24 ± 0.13 kg) were divided into 4 groups: the control group was gavaged with PBS, the Iturin A group was gavaged with 320 mg/kg BW Iturin A (Baishitai, Guangzhou, China), the STb-Rosetta group was gavaged with 1 × 10^10^ CFU/mL STb-R, and the Iturin A + STb-Rosetta group was gavaged with 320 mg/kg Iturin A + 1 × 10^10^ CFU/mL STb-R. Seven piglets in each group were fed 100 mL of these solutions for 10 days. Each piglet was fed in individual pens and one piglet is one replication. All piglets were allowed to access feed and water ad libitum in the pen; their basic diet was formulated to meet the National Research Council (NRC) 2012 requirements, ingredient composition and nutritional levels of the diets are provided in Appendix A. All piglets were sacrificed on the 11th day for sample collection.

### 2.2. Sample Collection

After 12 h fasting, all piglets (BW = 12.4 ± 0.21 kg) were slaughtered and the hypothalamus and pituitary were collected for total ribonucleic acid (RNA) extraction. Then, their skeletal muscles were completely isolated, and the weight was measured. The semimembranosus muscle was fast-frozen with liquid nitrogen, and all samples were stored at −80 °C.

### 2.3. RNA Extraction and cDNA Synthesis

Total RNA from the hypothalamus, pituitary, and semimembranosus muscle of piglets were extracted by TRIzol Reagent (Thermo Fisher Scientific, Carlsbad, CA, USA) and checked for integrity by 1.5% agarose gel electrophoresis. Then, double-stranded DNA in RNA sample were eliminated by gDNA Remover (EZBioscience, Roseville, CA, USA) and reacted at 25 °C for 5 min. After that, 4× RT Master Mix (EZBioscience, Roseville, CA, USA) and primers were added and reacted at 42 °C for 15 min to synthesize first-strand cDNA.

### 2.4. Quantitative Real-Time Polymerase Chain Reaction (qRT-PCR)

The mRNA abundance (*n* = 7) was determined by qRT-PCR using the QuantStudio 3 Quantitative Real-Time PCR system (Thermo Fisher Scientific, Waltham, MA, USA) and SYBR Green Real-Time PCR Master Mix (GenStar, Beijing, China). Specific primer pairs of growth-hormone-releasing hormone (GHRH), growth hormone (GH), growth-hormone-releasing hormone receptor (GHRHR), growth hormone receptor (GHR), insulin-like growth factor 1 (IGF-1), insulin-like growth factor 1 receptor (IGF-1R), glyceraldehyde-3-phosphate dehydrogenase (GAPDH), and β-actin were designed using Primer 5.0 software (Appendix A). A melting curve analysis was conducted to confirm the specificity of each product, and product sizes were verified in ethidium bromide-stained 1.5% agarose gels in Tris acetate-EDTA buffer. Quantitative data were obtained using the 2^−ΔCt^ method. The experiment was performed in triplicate.

### 2.5. Western Blotting

The semimembranosus muscle was collected for western blotting analysis as previously described [16]. Samples were homogenized with solution containing tissue lysis buffer (1% SDS, 8 mol/L urea) and 1 mmol/L phenylmethanesulfonyl fluoride (PMSF, Sigma-Aldrich, St. Louis, MO, USA). After protein concentrations were detected by micro-bicinchoninic acid assay kit (Thermo-Fisher, Waltham, MA, USA), the samples were treated with 5× loading buffer (GenStar, Beijing, China) and the proteins were separated by 10% sodium dodecyl sulfate polyacrylamide gel electrophoresis (SDS-PAGE). Then, the protein was incubated by polyvinylidene fluoride (PVDF) membranes (Millipore, Darmstadt, Germany) for primary antibody: Anti PI3K (CST-4257), Anti p-PI3K (Tyr458; CST-4228), Anti Akt (CST-9272), Anti p-Akt (Thr308, CST-13038), Anti p-mTOR (Ser2448; CST-5536), Anti mTOR (CST-2972), Anti p-S6K1 (Thr389; CST-9205), Anti S6K1 (CST-9202), Anti-p-S6 (Ser235/236; CST-2211), Anti S6 (CST-2217), Anti p-4EBP1 (Thr70; CST 9455), Anti 4EBP1 (CST-9452), Anti eIF4E (CST-2067), and Anti β-actin (CST-4970); these antibodies were purchased from Cell Signaling Technology (Beverly, MA, USA). Anti IGF1R (310158) was purchased from Zen Bioscience (Chengdu, China) and Anti IRS-1 (SC-8038) was purchased from Santa Cruz Biotechnology (Santa Cruz, CA, USA). Moreover, the second antibody (anti-rabbit IgG, BS13278; anti-mouse IgG, BS30503) was purchased from Bioworld Technology (St. Louis Park, MN, USA). Finally, the protein was detected by an enhanced chemiluminescence system (Protein Simple, San Jose, CA, USA) and analyzed by Image J analysis software.

### 2.6. Histology and Immunohistochemical Analysis

The semimembranosus muscle samples were embedded in Tissue-Tek to prepare cryosections (5 µm). The samples were observed by scanning electron microscopy and stained with hematoxylin-eosin (H&E) for histological analysis. Moreover, the tissue slides were incubated with 0.1% Triton X-100 for 10 min and 5% BSA for 30 min. Next, the proliferating cells were stained with the primary antibody Ki67 (NB500-170, Novus, Miami, FL, USA) for 12 h at 4 °C and the secondary antibody for 90 min at 25 °C. After the nuclei were stained with DAPI for 10 min, the tissue slides were photographed by immunofluorescence microscopy (Ti2-U, Nikon, Tokyo, Japan).

### 2.7. Statistical Analysis

The results were analyzed with SPSS software version (SPSS, Inc., Chicago, IL, USA). In brief, the data were statistically analyzed by one-way analysis of variance (ANOVA) and the Tukey’s test was used to put up multiple comparisons. All data are presented as the mean ± standard error (SEM), and differences were considered statistically significant when *p* < 0.05 and extremely significant when *p* < 0.01.

## 3. Results

### 3.1. Iturin A Rescued STb-R-Restricted Pork Skeletal Muscle Growth

As shown in Figure 1, STb-R significantly reduced piglet total weight gain (Figure 1A, *p* < 0.05), semimembranosus muscle weight (Figure 1B, *p* < 0.05), and semitendinosus muscle weight (Figure 1C, *p* < 0.05) compared to those of the control group, indicating that STb-R impeded skeletal muscle mass accumulation, leading to the growth inhibition of the piglets. Compared with the piglets in the STb-R group, the piglets in the Iturin A + STb-R group showed a significant increase in total weight gain (Figure 1A, *p* < 0.05) and semimembranosus muscle weight (Figure 1B, *p* < 0.05) but not semitendinosus muscle weight (Figure 1C, *p* > 0.05). Therefore, we selected the semimembranosus muscle for further study. In addition, there were slight but not significant changes in other skeletal muscles of the forequarters and hindquarters after the oral administration of Iturin A and STb-R (Appendix A, *p* > 0.05).

### 3.2. Iturin A Activated the Growth Axis to Rescue Pork Skeletal Growth

To explore the changes in the growth axis response to STb-R and Iturin A and their relationships with piglet skeletal muscle growth, we detected the relevant mRNA levels of the somatotropic axis. The results showed that compared with the control, Iturin A significantly increased the mRNA abundance of GHRH in the hypothalamus (Figure 2A, *p* < 0.05), GHRHR (Figure 2B, *p* < 0.05) and GH in the pituitary (Figure 2C, *p* < 0.01), GHR (Figure 2D, *p* < 0.05), IGF-1 (Figure 2E, *p* < 0.05), and IGF-1R (Figure 2F, *p* < 0.01) in the semimembranosus muscle. STb-R significantly inhibited the mRNA levels of the growth axis-related molecules GHRHR (Figure 2B, *p* < 0.05), GHR (Figure 2D, *p* < 0.01), and IGF-1R (Figure 1F, *p* < 0.05). Moreover, the mRNA abundance of the GHRHR (Figure 2B, *p* < 0.05), GH (Figure 2C, *p* < 0.01), GHR (Figure 2D, *p* < 0.05), IGF-1 (Figure 2E, *p* < 0.01), and IGF-1R (Figure 2D, *p* < 0.05) were significantly increased in the Iturin A + STb-R group compared with the STb-R group. In summary, our results suggest that Iturin A alleviated the inhibition of skeletal muscle growth in piglets by STb-R through the activation of the growth axis.

### 3.3. Iturin A Altered the Myofiber Characteristics in Semimembranosus Muscle

In addition, the results of H&E staining showed that STb-R significantly increased the mean fiber number (Figure 3A,B, *p* < 0.01) but decreased the mean fiber CSA of the semimembranosus muscle compared with those of the control group (Figure 3A,C, *p* < 0.01), indicating that the growth of the semimembranosus muscle was depressed after STb-R infection. Iturin A significantly decreased the mean fiber number (Figure 3A,B, *p* < 0.05) and increased the mean fiber CSA (Figure 3A,C, *p* < 0.01). After the gavage of Iturin A, both values recovered to the control levels (Figure 3A–C, *p* > 0.05). Accordingly, Iturin A improved the STb-restricted growth of the semimembranosus muscle.

### 3.4. Iturin A Rescued the Proliferation of Cells in Semimembranosus Muscle

To evaluate the proliferation of cells in semimembranosus muscle, we detected the proliferation marker Ki67 by immunohistochemical analysis. Based on the total cells (stained with DAPI), we found that STb-R significantly reduced the Ki67-positive cells compared with those of the control group, but the level was significantly increased in the Iturin A group (Figure 4A,B, *p* < 0.05). In addition, the relative mean number of STb-restricted Ki67-positive cells recovered to the control level in the Iturin A + STb-R group (Figure 4A,B). These data suggest that Iturin A improved the proliferation of cells in semimembranosus muscle.

### 3.5. Iturin A Reactivated the mTORC1 Pathway in Semimembranosus Muscle

As shown in Figure 5, STb-R significantly decreased the relative protein levels of IGF-1R and IRS-1 (Figure 5A,B, *p* < 0.01). Furthermore, the levels of key proteins in the PI3K/Akt/mTORC1 signaling pathway, including p-PI3K, p-Akt, p-mTOR, p-S6K1, p-4EBP1, and eIF4E, all significantly decreased after STb-R infection (Figure 5A,B, *p* < 0.05), which confirmed the inhibitory effect of STb-R on porcine skeletal muscle growth. In contrast, proteins other than p-Akt and p-4EBP1 were significantly increased by Iturin A compared with those of the control group. Notably, the expression of IGF-1R, IRS-1, p-PI3K, p-4EBP1, and Eif4e in the Iturin A + STb-R group recovered to normal levels (*p* > 0.05), and the protein level of p-Akt was significantly increased compared with that in the control group (Figure 5A,B, *p* < 0.05). These data demonstrate that the IGF-1-Akt-mTORC1 signaling pathway, which was restricted by STb-R, was recovered by Iturin A. Consequently, Iturin A rescued the growth of muscle by reactivating the I hypothalamic-pituitary-mTORC1 growth axis pathway in semimembranosus muscle (Figure 6).

## 4. Discussion

In animal science, protecting pigs against bacterial infection is a strategy to improve animal production [17]. Moreover, maintaining pork production to meet the increasing demand for animal protein is a major challenge after the prohibition of antibiotics [2]. AMPs have been regarded as the most promising ASs due to their broad-spectrum antimicrobial activity and low development of resistance [18]. Additionally, a review reported that some AMPs can improve the growth performance of piglets by protecting health and enhancing immunity [4]. Therefore, the development of new AMPs and the validation of their efficacy from multiple perspectives are necessary. Skeletal muscle comprises approximately 40% of total body weight and contains most body proteins [19]; it is also an important index for evaluating the efficiency of new AMPs. However, the relationship between skeletal muscle growth and Iturin A is still unknown.

In the current study, total skeletal muscle was isolated to assess the effect on piglet growth after the oral administration of Iturin A and STb-R. A substantial body of evidence has revealed that the performance of piglets is significantly decreased after challenge with enterotoxigenic *E. coli* (ETEC) [20,21], which is a common pathogen in animal production [22]. Additionally, as shown in our study, the final body weight and total weight gain were significantly suppressed by STb-R. In addition, the reduction in semimembranosus and semitendinosus mass indicated that the skeletal muscle growth of piglets was inhibited, although the other muscle masses were reduced slightly. Furthermore, relevant studies on AMPs, such as cecropin AD [23], have shown that they protect animal health and improve the performance of both healthy and challenged piglets. Although Iturin A rescued the growth of piglets challenged by STb-R in our study, the growth-promoting effect on healthy piglets was not significant, probably due to the short duration of administration, and the effect of long-term administration of Iturin A remains to be studied. A similar study reported that Iturin A administrated for 15 d significantly inhibited tumor growth; however, there were no significant effects on body weight, food intake, and viscera index in mice [24]. Taken together, Iturin A maybe better at resisting damaging stimulations.

The hypothalamus-pituitary growth axis is required for a number of biological processes, including growth and metabolism [12]. Due to its central and positive effects of the GH/IGF-1 system on muscle mass, its regulation in response to Iturin A is of interest. In the present study, we observed an increase in the mRNA abundance of growth axis-related genes induced by Iturin A, indicating that Iturin A activates the growth axis to improve muscle growth. In contrast, STb-R decreased the expression of GHRHR, GHR, and IGF-1R, proving its inhibitory effect on muscle growth, which may be due to the following two reasons. On the one hand, STb-R invasion leads to inflammation, which modifies the secretion of hormones associated with the growth axis, since the concentration of IGF-1 and its synthesis in the liver decrease shortly after acute infection or endotoxin administration [25,26]. On the other hand, STb-R infection impedes nutrient absorption, thus decreasing the expression of the growth axis, and the destruction of the intestinal structure when challenged by ETEC also supports this view [21,27]. In general, our results demonstrate that Iturin A relieves the inhibitory effect of STb-R on the growth axis.

Moreover, we found that Iturin A rescued the STb-R-restricted muscle fiber CSA of the semimembranosus muscle. The increased CSA of muscle fiber means myofiber hypertrophy, and this reflected enhanced protein synthesis ability. In contrast, the elevated myofiber size may result in decreased myofiber number. Additionally, the proliferating cells in skeletal muscle are most likely satellite cells responsible for postnatal skeletal muscle growth [28], and their proliferative capacity was improved by Iturin A. Additionally, as shown in a previous study, the upregulation of IGF-1 expression leads to skeletal muscle hypertrophy and the activation of the downstream mTORC1 pathway [29,30]. However, mice lacking IGF-1R or GHR exhibited a reduced myofiber number and area, with accompanying functional deficits [31] Furthermore, our lab has reported that the mTORC1 pathway participates in not only the regulation of satellite cell proliferation and differentiation, but also the synthesis and deposition of protein in skeletal muscle [15]. Consequently, the improvement in skeletal muscle by Iturin A is closely associated with the IGF-1-mTORC1 signaling cascade.

mTORC1 signaling is a highly conserved serine/threonine kinase pathway that regulates metabolism and cell growth in response to multiple environmental cues, including nutrients and hormones [32,33]. In the present study, the elevated protein levels of IGF-1R and IRS-1 further proved the activation of the growth axis by Iturin A, and consistent with the change in the growth axis in response to Iturin A and STb-R, we found that Iturin A strongly enhanced the STb-R-restricted expression of mTORC1 signaling pathways, such as p-mTOR, p-S6K1, and eif4E. Their elevation was also observed in lysine-induced porcine myofiber hypertrophy [14,15]. Compared with the robust elevation in mTORC1, we observed that the promoting effect of Iturin A on skeletal muscle mass was not significant, although it was sufficient to rescue the growth depressed by STb-R infection. This finding may be explained by the fact that the activation of mTORC1 signaling by Iturin A requires prolonged feeding to deposit proteins and to significantly increase skeletal muscle weight, but excellent protection has already been shown in the weaning stage of piglets to resist pathogenic bacterial invasion.

## 5. Conclusions

In the current study, our results showed that STb-R-caused porcine skeletal muscle growth restriction was rescued by Iturin A via the activation of the hypothalamus-pituitary-IGF-1 growth axis, which increased the mTORC1 pathway in skeletal muscle. Moreover, the proliferation of muscle cells and the fiber CSA were improved by Iturin A.

## Figures and Tables

**Figure 1 animals-12-01568-f001:**
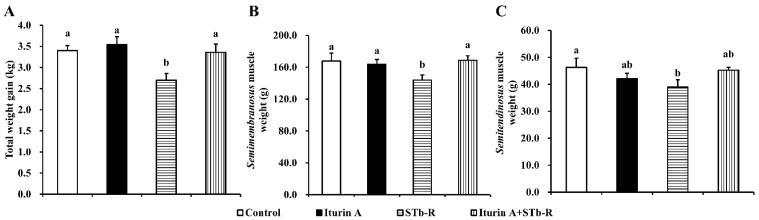
Effects of Iturin A on STb-R-restricted porcine skeletal muscle growth: (**A**) The total weight gain (kg) of piglets in each group after been fed for 10 days; (**B**) the weight (g) of semimembranosus muscle of piglets in different groups; and (**C**) the weight (g) of the semitendinosus muscle of piglets in different groups. Data are presented as mean ± SEM, and the different lowercase letters above standard error bars indicate significant differences, *n* = 7.

**Figure 2 animals-12-01568-f002:**
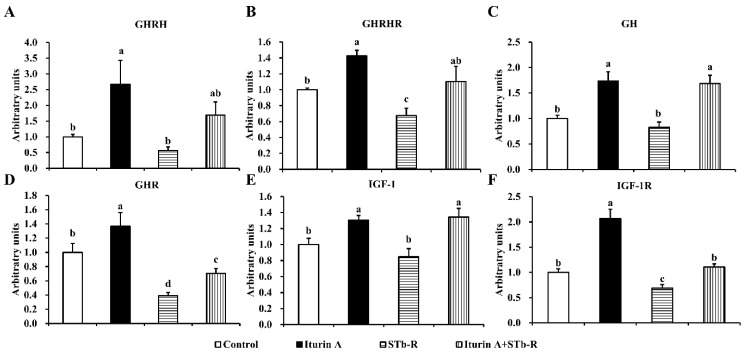
Effects of Iturin A on STb-R-suppressed relative mRNA abundance of growth axis in hypothalamus, pituitary, and muscle. qRT-PCR validation of the mRNA abundance of GHRH (**A**) in the hypothalamus, GHRHR (**B**) and GH (**C**) in the pituitary, GHR (**D**), IGF-1(**E**), and IGF-1R (**F**) in the semimembranosus muscle, all of them normalized to GAPDH and shown relative to the control group. Data are presented as mean ± SEM, and the different lowercase letters above standard error bars indicate significant differences, *n* = 7.

**Figure 3 animals-12-01568-f003:**
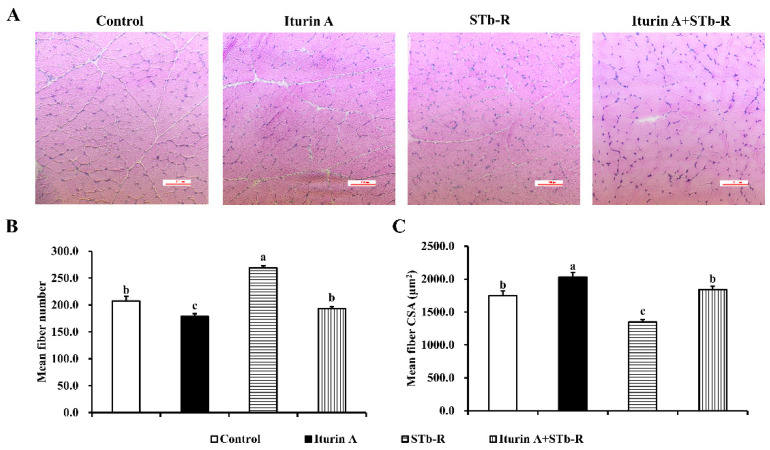
Effects of Iturin A on STb-R-reversed myofiber number and area in semimembranosus muscle. (**A**) Hematoxylin-Eosin (H&E) staining was used to confirm the promoted effect of Iturin A on semimembranosus muscle growth, bar 200×. (**B**,**C**) Quantification of the relative fiber number (**B**) and relative mean fiber cross-sectional area (CSA) (**C**) in semimembranosus muscle. Data are presented as mean ± SEM, and the different lowercase letters above standard error bars indicate significant differences, *n* = 3.

**Figure 4 animals-12-01568-f004:**
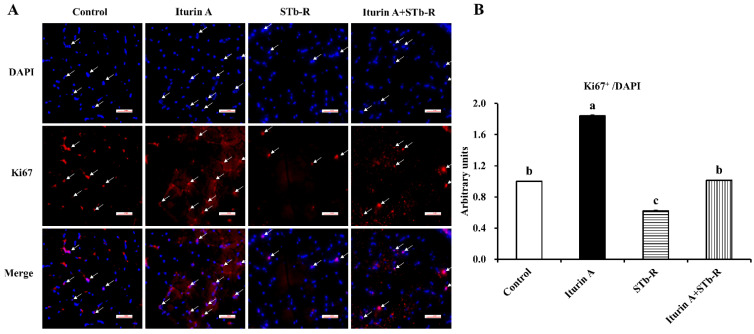
Effects of Iturin A on STb-R-inhibited proliferated cell numbers in semimembranosus muscle. (**A**) Immunofluorescence staining with Ki67 was performed to confirm the changes in proliferating cells in semimembranosus muscle after being gavaged with Iturin A and STb-R. The arrows presented DAPI/Ki67 targeted cells. Bar 200×. (**B**) Quantification of Ki67-positive cells (red) based on total cells (staining with DAPI, blue) in semimembranosus muscle. Data was presented as mean ± SEM, and the different lowercase letters above standard error bars indicate significant differences, *n* = 3.

**Figure 5 animals-12-01568-f005:**
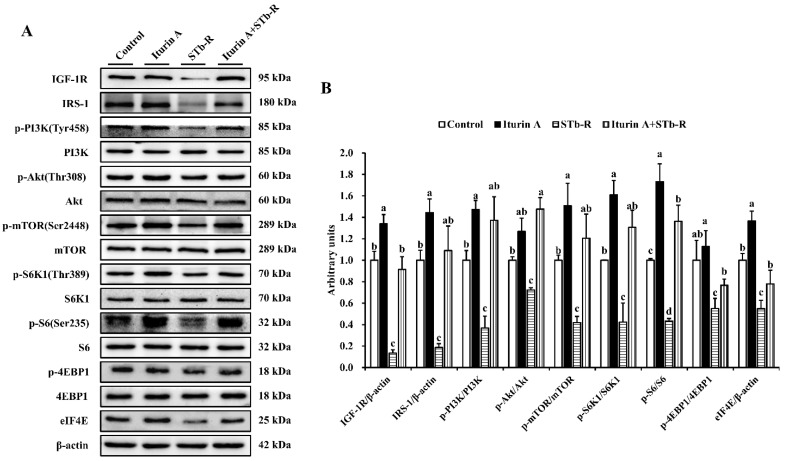
Effects of Iturin A on STb-R-restricted mTORC1 pathway in semimembranosus muscle. (**A**) Western blotting analysis of IGF-1R, IRS-1, p-PI3K, p-Akt, p-mTOR, p-S6K1, p-S6, p-4EBP1, and eIF4E in semimembranosus muscle shows the change of mTORC1 signaling pathway after being fed with Iturin A and STb-R. (**B**) Quantification of the western blotting results and the values of the relative protein level are represented as the ratio of IGF-1R/β-actin, IRS-1/β-actin, p-PI3K/PI3K, p-Akt/Akt, p-mTOR/mTOR, p-S6K1/S6K1, p-S6/S6, p-4EBP1/4EBP1, and eIF4E/β-actin. Data are presented as mean ± SEM, and the different lowercase letters above standard error bars indicate significant differences, *n* = 3.

**Figure 6 animals-12-01568-f006:**
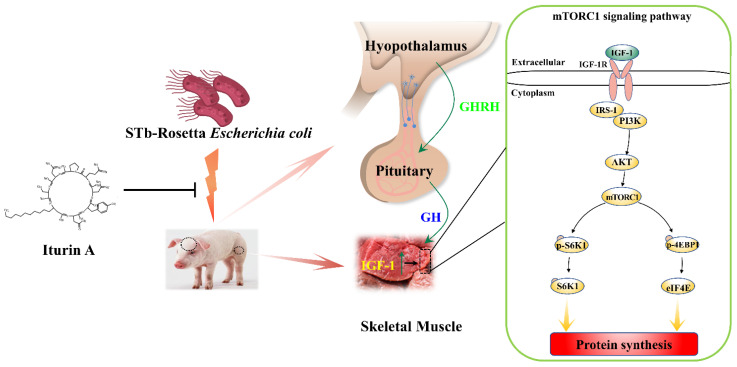
Iturin A rescued STb-R-induced pork skeletal muscle growth restriction through hypothalamic-pituitary-mTORC1 growth axis. In detail, the engineered STb-Rosetta Escherichia coli (STb-R) inhibited IGF-1 expression in muscle via the hypothalamic-pituitary growth axis, and the downstream mTORC1 signaling pathway responded to change in IGF-1, ultimately limiting skeletal muscle growth in piglets. Iturin A reactivated the hypothalamic-pituitary-mTORC1 growth axis, which promoted skeletal muscle growth and alleviated STb-R restriction in piglets.

## Data Availability

The data presented in this study are available on request from the corresponding author.

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
