# Peer review of "Iturin A Rescued STb-R-Induced Pork Skeletal Muscle Growth Restriction through the Hypothalamic-Pituitary-mTORC1 Growth Axis"

_animals, 2022, doi:10.3390/ani12121568_

Round 1
Reviewer 1 Report
The authors investigated the effects of Iturin A on skeletal muscle growth of weaned piglets infected by STb-Rosetta Escherichia coli (STb-R). They indicated that the hypothalamic-pituitary growth axis mediated Iturin A reactivated the mTORC1 pathway to rescue STb-R-restricted pork skeletal muscle growth.
The manuscript is well-written, but some modification is required before the publication.
My major comments are the following:
- Materials and methods section:
- Wester blotting technique was not described in particular; for example in lane 106: “Then, the protein was incubated…..”what protein are we talking about? Do you mean all proteins extracted from muscle? How did you separate proteins before western blotting? By electhrophoresis? Between primary antibody, please check if you indicated all antibody (I can not find anti beta-actin and anti S/6). Moreover you should report how relative protein levels are expressed, explaining in particular the ratios reported in fig.5B.
- 2- Statistical analysis paragraph should be rewritten: in lane 130: “the results were statistically analysed using using Tukey’s test, and the remaining results were analysed by one-way analysis of variance (ANOVA)”; I can’t understand what data were analysed by Tukey’s test and what by ANOVA. Why did you use two different tests? Did you analysed the distribution of your data before choosing the test? Did you apply a homoskedasticity test of the sample in order to verify that the body weight of the piglet at the beginning was homogeneus?
- Results section:
- all results statistically significant presented a p<0.05, no results have differences with p<0.01, although in charts differences seems sometimes extremely significant. Please check if p is always <0.05.
- in fig. 1 histogram chart A and histogram chart B report the same results; you could just report chart B.
- Discussion: I really appreciate that you have highlighted the limits of your study due probably to the short duration of administration of Iturin A (lanes 255-259). Do you know similar studies where Iturin A was administrated for more time? If yes, can you report the references?
Author Response
To Reviewer 1:
Thank you very much indeed for your excellent comments on our manuscript. We are very glad to revise our paper according to your comments. The changes of revised manuscript were highlighted in yellow, and the explanations for comments were listed as below.
Question 1: Western blotting technique was not described in particular; for example in lane 106: “Then, the protein was incubated…..” what protein are we talking about? Do you mean all proteins extracted from muscle? How did you separate proteins before western blotting? By electhrophoresis? Between primary antibody, please check if you indicated all antibody (I can not find anti beta-actin and anti S/6). Moreover, you should report how relative protein levels are expressed, explaining in particular the ratios reported in fig.5B.
Answer: Sorry, this is our carelessness. The proteins detected by western blotting were all extracted from semimembranosus muscle. Before western blotting, total proteins (10 µg) were separated through 10% sodium dodecyl sulfate polyacrylamide gel electrophoresis (SDS-PAGE). Anti S6 antibody (CST‐2217) and anti β-actin antibody (CST-4970) were also purchased from Cell Signaling Technology (Beverly, MA, USA). The definition of relative protein levels is explained in the legend of Figure 5B. These details are supplemented in the revised Manuscript, please see Page 3, Line 113-122 and Page 7, Line 219-222.
Question 2: Statistical analysis paragraph should be rewritten: in lane 130: “the results were statistically analyzed using Tukey’s test, and the remaining results were analysed by one-way analysis of variance (ANOVA)”; I can’t understand what data were analysed by Tukey’s test and what by ANOVA. Why did you use two different tests? Did you analysed the distribution of your data before choosing the test? Did you apply a homoskedasticity test of the sample in order to verify that the body weight of the piglet at the beginning was homogeneus?
Answer: All data were analyzed by ANOVA and the Tukey’s test was used to put up multiple comparison. We have justified the distribution of our data. In generally, it is unacceptable for us once the CV% beyond 15%. The homoskedasticity test was also applied before the beginning and the significance value p=0.571 > 0.05, indicating that the variance among all treatment groups is homogenous. We have re-written the statistical analysis paragraph, please see Page 3-4, Line 140-144. Thank you for your reminding.
Question 3: All results statistically significant presented a p<0.05, no results have differences with p<0.01, although in charts differences seems sometimes extremely significant. Please check if p is always p<0.05.
Answer: Thank you for your notice. We have checked the data and revised the Manuscript. Please see Page 4, Line 169, 170, 172, 174; Page 5, Line 187, 189, 192; Page 8, Line 226.
Question 4: In fig. 1 histogram chart A and histogram chart B report the same results; you could just report chart B.
Answer: Thank you for your suggestion. We have removed the Figure 1A, please see the revised Figure 1 on Page 4, Line 147-161.
Question 5: Discussion: I really appreciate that you have highlighted the limits of your study due probably to the short duration of administration of Iturin A (lanes 255-259). Do you know similar studies where Iturin A was administrated for more time? If yes, can you report the references?
Answer: Yes, we have searched some studies about Iturin A administrated for more time. Zhao et al., (2021) reported that Iturin A administrated for 15 d significantly inhibited tumor growth, however, there were no significant effects on body weight, food intake, and viscera index in mice. Taken together, Iturin A maybe better at resisting damaging stimulations. We have discussed this on Page 9, Line 269-272.
Reference 24: Zhao, H.; Yan, L.; Guo, L.; Sun, H.; Huang, Q.; Shao, D.; Jiang, C.; Shi, J. Effects of Bacillus subtilis iturin A on HepG2 cells in vitro and vivo. AMB Express. 2021, 11(1), 67.
Reviewer 2 Report
Dear Authors,
the study is well described and the experimental design appropriated.
However, more information of piglets is required. Where the animals siblings and which feed was supplied and in which way (composition, nutritive content, distribution and intake).
The genes analyzed fro qRT-PCR are not reported in the section of material and methods, where also the housekeeping genes must be included. Please consider that only 1 hk gene (GAPDH) is not recognised as a standard method and at least 2, better 3, HK genes are needed for relative quantification. Being a relative quantification - this is what it appears - more details on the 2-DCt method are required.
Please, comment why the same proteins were not qunatified with qRT-PCR in the muscle.
Author Response
To Reviewer 2:
Thank you for your suggestion. We are pleased to revise our paper according to your comments. The changes of revised manuscript were highlighted in green, and the explanations for comments were listed as below.
Question 1: However, more information of piglets is required. Where the animals siblings and which feed was supplied and in which way (composition, nutritive content, distribution and intake).
Answer: Thank you for your reminding. The animals used in this study were half sibs (same semen, different sow). Each piglet was feed in individually pen and one piglet is one replication. All piglets were allowed to access feed and water ad libitum in pen, the basic diet was formulated to meet the National Research Council (NRC) 2012 requirements, ingredient composition and nutritional levels of the diets were provided in the Supplementary Table 1. Please see Page 2, Line 75-83.
Question 2: The genes analyzed for qRT-PCR are not reported in the section of material and methods, where also the housekeeping genes must be included. Please consider that only 1 hk gene (GAPDH) is not recognized as a standard method and at least 2, better 3, HK genes are needed for relative quantification. Being a relative quantification - this is what it appears - more details on the 2-DCt method are required.
Answer: Thank you for your suggestion. We have supplemented the details about genes in the material and methods section. Please see Page 3, Line 101-105. Meanwhile, we repeated this result with 2 housekeeping genes (GAPDH and β-actin). Please see Page 4, Line 167-176 and revised Figure 2. Thank you.
Question 3: Please, comment why the same proteins were not quantified with qRT-PCR in the muscle.
Answer: In our opinion, the proteins are the basic elementary component of life, while they are responsible for carrying through the functions of body cell. Skeletal muscle is terminally differentiated tissue, detecting the protein levels is enough to reflect the mechanism. Meanwhile, different from the genes in the hypothalamic-pituitary growth axis, the changes of mTORC1 pathway were always measured by the key protein levels. Actually, we would like to measure the same protein level of GHRH in the hypothalamus, GHRHR and GH in the pituitary, however, the hypothalamus and pituitary are too small, while the samples are too little to be measured by western blotting. Thank you for your suggestion.
Round 2
Reviewer 1 Report
The authors revised the last version of the manuscript according the suggestion; this manuscript should be now suitable for publication in present form.
Author Response
To Reviewer 1:
Thank you for your recommendation. As substitute of antibiotic, antimicrobial peptide is the hotspot in the research of animals nutrition currently. We would like to screen effective antibiotic substitutes and investigate further mechanisms. And then, produce healthy animal products.